# Sleep Quality between Nurses and the General Population during the COVID-19 Pandemic in Portugal: What Are the Differences?

**DOI:** 10.3390/ijerph20085531

**Published:** 2023-04-17

**Authors:** Francisco Sampaio, Susana Gaspar, César Fonseca, Manuel José Lopes, Teresa Paiva, Lara Guedes de Pinho

**Affiliations:** 1Nursing School of Porto, Rua Dr. António Bernardino de Almeida, 830, 844, 856, 4200-072 Porto, Portugal; 2CINTESIS@RISE, Nursing School of Porto (ESEP), Rua Dr. Plácido da Costa, 4200-450 Porto, Portugal; 3School of Health, Polytechnic Institute of Beja, R. Dr. José Correia Maltez, 7800-111 Beja, Portugal; 4Nursing Department, Universidade de Évora, Largo do Senhor da Pobreza, 7000-811 Évora, Portugal; 5Comprehensive Health Research Centre (CHRC), Universidade de Évora, Largo do Senhor da Pobreza, 7000-811 Évora, Portugal; 6CENC—Sleep Medicine Center, Rua Conde das Antas, 5, 1070-068 Lisboa, Portugal; teresapaiva0@gmail.com; 7Instituto de Saúde Ambiental, Faculdade de Medicina, Universidade de Lisboa, Av. Prof. Egas Moniz, Ed. Egas Moniz, Piso 0, Ala C, 1649-026 Lisboa, Portugal; 8Comprehensive Health Research Center, Nova Medical School, Universidade Nova de Lisboa, Rua do Instituto Bacteriológico, 5, 1150-082 Lisboa, Portugal

**Keywords:** sleep quality, mental health, nurses, population, COVID-19, pandemics, Portugal, cross-sectional studies

## Abstract

Although several studies have described the impact of the COVID-19 pandemic, particularly on sleep quality, there are few studies that, in the same time period and using the same assessment tools, compare sleep quality and mental health status between nurses and the general population. Thus, the aim of this study was to (a) examine whether there were differences between nurses and the general population regarding sleep quality and mental health status during the COVID-19 pandemic and (b) identify which factors may explain sleep quality during the COVID-19 pandemic. To do that, we carried out a cross-sectional study in Portugal. Data were collected using an online survey platform during the first COVID-19 wave, from April to August 2020. Nurses presented poorer sleep quality than the general population, as well as higher anxiety levels. Irritability and worries about the future were two of the factors that might explain those differences. Thus, we can conclude that irritability and worries about the future are dimensions of anxiety that were associated with poor sleep quality during the COVID-19 pandemic. Thus, it would be important to adopt regular anxiety and sleep assessments, particularly for nurses, and to implement strategies to reduce this problem.

## 1. Introduction

The pandemic caused by coronavirus disease (COVID-19) has been a stressful life event. Soon after the COVID-19 crisis was declared a pandemic, significant concerns about people’s mental health were raised [1,2]. Several studies conducted during the outbreak of the COVID-19 pandemic reported that many people considered their mental health to have worsened [3,4]. Nevertheless, a systematic review and meta-analysis of longitudinal cohort studies examining changes in mental health among the same group of participants before versus during the COVID-19 pandemic in 2020 concluded that there was a slight increase in mental health symptoms after the outbreak of the COVID-19 pandemic (March–April 2020) that declined afterward (May–July 2020) and was comparable to pre-pandemic levels among most population subgroups and symptom types [5].

Additionally, at the very beginning of the COVID-19 pandemic, many concerns regarding people’s sleep quality were raised [6,7]. Several studies published between December 2019 and November 2021 confirmed a trend toward sleep disruption during the COVID-19 pandemic [7,8,9,10,11]. A systematic review, meta-analysis and meta-regression published in 2022 confirmed this trend, in which overall impaired sleep and widespread subthreshold insomnia during the COVID-19 pandemic were found [12]. Another systematic review, meta-analysis and meta-regression published in 2022 pointed out that the estimated prevalence of sleep problems was 52.39% among patients infected with COVID-19, 45.96% among children and adolescents, 42.47% among healthcare workers, 41.50% among special populations with healthcare needs, 41.16% among university students, and 36.73% among the general population [13].

The results of the studies that have been carried out in Portugal tend to be in line with these results. Regarding the mental health of the general population during the first lockdown, severe depression, anxiety and stress symptoms were found in 7.6%, 9.1% and 9.3%, respectively, of a sample of 1280 Portuguese individuals. In the same sample, severe obsessive-compulsive symptoms were present in 12.4% of the participants [14]. However, in a longitudinal study also carried out in the general population, depression, anxiety and stress symptoms seemed to improve over time, while the perception of quality of life and sleep worsened [15].

The studies carried out in Portugal that involved nurses who, according to the literature, were at increased risk of having psychological problems compared to other healthcare workers during the COVID-19 pandemic [16], also presented similar findings when compared to the studies involving the general population. Thus, nurses presented higher depression, anxiety and stress levels than subjects from the general population during the first lockdown, suggesting that nurses’ mental health status seemed to be affected at that moment [17]. Nevertheless, a longitudinal study indicated that nurses’ sleep quality and symptoms of depression, anxiety and stress presented a positive variation over the COVID-19 pandemic, suggesting that a psychological adaptation phenomenon was taking place [18]. Contrary to the general population in Portugal, who reported a decline in the perception of sleep quality throughout the COVID-19 pandemic [15], nurses reported that their sleep quality tended to improve over time [18]. Thus, while in the general population the perception of sleep quality presented an initial decrease (between 23 March–5 April 2020 and 13 April–26 April 2020) with a later slight increase that did not reach significance (between 2 May and 15 May 2020) [15], the number of nurses who reported a poor sleep quality between 31 March and 6 April 2020 significantly decreased over time (until 4 May 2020) [18].

Although several studies aiming to assess the sleep quality of nurses and the general population throughout the COVID-19 pandemic have been conducted, few studies have compared the sleep quality of nurses and the general population in the same timeframe and using the same measures. In Portugal, some studies were carried out concerning sleep habits and sleep quality during the COVID-19 pandemic, such as those conducted by Silva and Sobral [19] and by Paiva et al. [20]. However, none of these allowed us to compare, in a large sample and during the same timeframe, the sleep quality of nurses and the general population, which, according to the findings of the abovementioned studies [15,17], seemed to evolve differently throughout the COVID-19 pandemic.

Several factors may affect individuals’ sleep quality. Some of those factors are related to their mental health, as the correlation between sleep and mental health has been shown in recent years [21,22]. The positive correlation between sleep disturbances and mental health problems was confirmed by a systematic review and meta-analysis of studies involving healthcare workers conducted during the COVID-19 pandemic [23]. Nonetheless, it is relevant to note that, according to that systematic review [23], some conditions faced by healthcare workers during the COVID-19 pandemic, such as exposition to the disease, the possible transmission of infection to family members, shortage of personal protective equipment, extended working hours, and decisions about allocating limited resources to patients, also presented a risk of adverse mental health. Although definitive conclusions regarding bidirectionality cannot be made for most sleep disturbances, the best available evidence suggests that some of the disturbances, such as insomnia, are bidirectionally related to mental health problems such as anxiety and depression [24].

Although some studies have been conducted during the COVID-19 pandemic to examine the connections between sleep quality and mental health [25,26], none have clearly explored the factors that may explain sleep quality during the pandemic. In Portugal, the study carried out by Paiva et al. [20] identified differences between two groups (poor and good sleep quality) for mood and economic problems. However, no regression analysis was carried out to better explain how those variables may influence sleep quality nor were subgroups (nurses or general population) considered as variables that may influence sleep quality.

Thus, this study aimed to (a) examine whether there were differences between nurses and the general population regarding sleep quality and mental health status during the COVID-19 pandemic and (b) identify which factors may explain sleep quality during the COVID-19 pandemic.

## 2. Materials and Methods

### 2.1. Study Design and Procedures

The present study is part of a larger project, ‘Sleep, COVID and Habits,’ approved by CENC—Sleep Medicine Center Ethical Committee (ref. no. 1/2020), for which there was no funding, public or private. This is a cross-sectional study and data were collected using a questionnaire based on an online survey platform, Survey Legend^®^. The use of this platform allowed for data analysis and statistical use. The questionnaire was developed by a team of experts in the field using measures based on the most important dimensions in the study. Afterwards, a team of peers reviewed the questionnaire and made suggestions for improvement. Lastly, before implementing the survey, the questionnaire was applied to a subsample for checking understanding and relevance of each question. The consent of the participants was obtained before starting to fill out the questionnaire. Additionally, before starting the questionnaire and asking for consent, the objectives and procedures of the study were presented to each participant. To preserve confidentiality of the data, surveys were conducted anonymously.

Data collection was carried out online during the first COVID-19 wave, from April to August 2020. The questionnaires were distributed through contacts of health care professional associations, healthcare institutions, and social media for the general population. The surveys were employed to obtain data from the Portuguese mainland and islands (Madeira and Azores). Although the surveys were adapted to each main subgroup, they all had a core and common structure used in this study. Further details of the methodologic procedures used in the ‘Sleep, COVID and Habits’ project can be accessed elsewhere [20,27,28,29]. Inclusion criteria were: adults aged 18 or older, with residence in Portugal mainland or islands, who belonged to a subgroup such as the general population, sleep disorder patients, COVID-involved professionals (medical doctors and nurses), and COVID-affected professionals (teachers, psychologists, and dentists) [20]. A convenience sampling method was adopted in this study.

### 2.2. Participants

The participants were 5479 adults (aged ≥18 years) divided into subgroups. The present study only analyzed the general population (*n* = 747) and nurses (*n* = 629) subgroups, which formed a total of 1376 participants with an average age of 45.64 ± 12.26 years; 80.20% (*n* = 1103) were female (Table 1).

### 2.3. Measures

For this study, several measures related to sociodemographic characteristics (sex, age, civil status, and education level) were considered (Table 1) to analyze sleep quality. Table 2 describes all variables used in this study. The variables were obtained by the answers to the following questions: (1) Sleep quality variable: ‘How do you rate the quality of your sleep during the COVID-19 pandemic?’ (1 = minimum sleep quality; 10 = maximum sleep quality); (2) Sleep awakening quality variable: ‘How do you rate the quality of your awakening during the COVID-19 pandemic?’ (1 = minimum sleep awakening quality; 10 = maximum sleep awakening quality); (3) Sleep duration weekdays variable: ‘How many real hours of sleep, on average, did you have daily during weekdays throughout the COVID-19 pandemic?’; (4) Sleep latency variable: ‘How long did it take to fall asleep in weekdays during the COVID-19 pandemic (in minutes)?’; (5) Depressive status variable: ‘On a scale of 1 (not depressed at all) to 10 (very depressed), how would you rate your depressive symptoms during the COVID-19 pandemic?’; (6) Irritability variable: ‘On a scale of 1 (not at all anxious) to 10 (very anxious), how would you rate your level of anxiety during the COVID-19 pandemic?’; (7) Economic problems variable: ‘On a scale of 1 (no problems) to 10 (severe problems), how would you rate your economic problems during the COVID-19 pandemic?’; and (8) Worries about the future variable: ‘On a scale of 1 (no worries) to 10 (huge concerns), how would you rate your worries about the uncertainty of the future during the COVID-19 pandemic?’ The ranges of all variables included in the study are described in Table 2.

### 2.4. Statistical Data Analysis

Data were analyzed using SPSS version 28 for Windows. Descriptive statistics (including mean, standard deviation, percentage) were calculated to characterize the participants. The chi-square test was used to analyze the relationship between the subgroups (nurses and general population) and sex in nominal variables. Independent sample *t* tests were used to analyze differences between the subgroups (nurses and general population) regarding age, sleep quality, sleep duration weekdays, sleep latency, sleep awakening quality, depressive symptoms, anxiety, irritability, economic problems, and worries about the future (quantitative variables).

Logistic regression models were conducted using the “Enter” method to analyze the relationship between having above-average sleep quality and subgroups (nurses and general population), sleep-related variables (sleep duration weekdays, and sleep latency), and well-being-related variables (depressive symptoms, anxiety, irritability, economic problems, and worries about the future). The ‘above average’ method was used because there is no defined cut-off value for sleep quality. This method is often used in other studies [30,31]. Two models were used: 1) unadjusted model and 2) adjusted model. The adjusted model was controlled for age and sex. The significance level was set at *p* < 0.05.

## 3. Results

Table 1 presents the sociodemographic characteristics of the participants. The overall sample had a mean age of 45.64 ± 12.26 years, with 80.2% (*n* = 1103) being female. From the overall sample, 45.70% were nurses (*n* = 629).

Several statistically significant differences between the nurse sample and general population sample were identified, as reported in Table 3. When compared with the general population, nurses reported (on average) lower sleep quality (with a higher percentage below the average), lower sleep awakening quality, higher anxiety, and higher depressive symptoms.

The results of the logistic regression analysis for having above average sleep quality during the first COVID-19 wave, according to the sleep-related variables and well-being-related variables with and without age and sex adjustment, are shown in Table 4. Both models are significant (*p* < 0.001) and explain approximately 23% and 21% of the models with and without adjustment, respectively.

In the unadjusted model, being a nurse decreased the likelihood of having an above average sleep quality by 42% (95% CI: 0.44, 0.77) compared to being a member of the general population. Additionally, greater depressive symptomatology (OR = 0.88; 95% CI: 0.80, 0.96) and worrying about the future (OR = 0.90; 95% CI: 0.84, 0.96) lowered the odds of having above-average sleep quality.

For the adjusted model, being a nurse decreased the likelihood of having above average sleep quality by 49% (95% CI: 0.38, 0.69) compared to being a member of the general population. Irritability emerged as a significant association, as increasing one value in the irritability score decreased the likelihood of having above average sleep quality by 9% (95% CI: 0.83, 0.99). Furthermore, worrying about the future remained a significant association, as increasing one value in the irritability score decreased the likelihood of having above average sleep quality by 11% (95% CI: 0.84, 0.96). However, significant depressive symptoms declined.

## 4. Discussion

Focusing on the purpose of this study, we found that 62.96% of nurses and 46.45% of the general population had poor sleep quality during the first COVID-19 wave, and that nurses had poorer sleep quality than the general population. These data are congruent with several studies involving meta-analyses [12,13,32,33]. Our study was conducted during the confinement period, so it is important to note that sleep disturbances were higher during confinement compared to nonconfinement, 42.49% versus 37.97% [13].

A meta-analysis conducted in 2020 (one of the first studies during the pandemic) concluded that the prevalence rate of psychological morbidities during the pandemic was highest in poor sleep quality (40%), followed by stress (34%), psychological distress (34%), insomnia (30%), post-traumatic stress symptoms (27%), anxiety (26%) and depression (26%) [32]. Additionally, when authors compared the burden of these psychological morbidities, they concluded that it was higher in healthcare workers than in the general population [32]. A meta-analysis conducted in 2021 showed that the estimated prevalence of sleep problems was higher in healthcare professionals (31%) than in the general population (18%) [33]. A recent meta-analysis (2022) came to the same conclusion: the estimated prevalence of sleep problems was higher among healthcare workers (42.47% [37.95%–47.12%]) than the general population (36.73% [32.32%–41.38%]) [13]. It is important to note that for the healthcare worker population, being a nurse was not a moderator of sleep disturbances during the COVID-19 pandemic [13]. Therefore, we can infer that the results are similar among health professionals. In addition, the same research showed that poor sleep quality was the main problem and explained 52% of the variance in the data [13]. None of these meta-analyses address the Portuguese context, but we find that the results of our study trend in the same direction. Although our results correspond to the beginning of the pandemic in Portugal (2020), by analyzing the evolution of the different meta-analyses carried out in different countries, we can conclude that the tendency is not toward stabilizing sleep quality, and this problem is expected to continue. In addition to issues related to the pandemic, there are other factors that may be associated with poor sleep quality in nurses, such as shift work that does not allow for regular circadian rhythm. It is therefore worth emphasizing the importance of preventing these problems in nurses with the implementation of appropriate interventions.

In our study, poorer sleep quality was associated with irritability when adjusted for age and sex. In fact, one of the consequences of poor sleep quality reported in the literature is irritability [34], so the results of this study are in line with these results. We cannot infer from our results whether irritability is a consequence or cause of poor sleep quality; however, it is likely to be a consequence according to the literature. A study of nursing students during the pandemic concluded that fear of COVID-19 was associated with their high irritability and poor sleep quality [35]. A meta-analysis also concluded that there was a significant association between sleep quality and irritability [36]

Another result of our study is the association between poor sleep quality and worries about the future. Other studies have shown similar results, indicating that the greater the worry about the future, the worse the sleep quality [37,38,39]. Irritability and worries about the future are also dimensions of anxiety [40]. There are several studies that associate anxiety with poor sleep quality [7,8,41,42,43]. One study of 628 healthcare professionals (81.4% were nurses), concluded that one of the factors associated with a higher risk of anxiety was poor sleep quality [42]. A study conducted in the United States found that participants who reported more COVID-related stress had more anxious symptoms and poorer sleep quality [25]. In our study, nurses had more anxiety than the general population, which is consistent with other Portuguese studies, and may, in part, explain these results. Therefore, it is important to promote nurses’ mental health through strategies such as raising awareness of the importance of practicing physical activities, relaxation and a healthy diet, taking breaks between work shifts, and expressing feelings/emotions [44,45]. This may be a factor that is also related to poor sleep quality, although in the logistic regression, the result was not significant. This may be due to the way the question was asked, since an evaluation scale was not used.

This study has limitations, such as the cross-sectional design. The data collection method was online, so possible bias in the selection of participants exists. As strengths, the sample size and the national coverage make this a representative study.

## 5. Conclusions

In Portugal, during the first wave of the COVID-19 pandemic, we found that nurses had poorer sleep quality than the general population and higher anxiety levels. Irritability and worries about the future are two of the factors that may explain these differences. Since both dimensions are characteristic of anxiety, it is important to adopt regular anxiety and sleep assessment measures for nurses and implement strategies to reduce this problem.

## Figures and Tables

**Table 1 ijerph-20-05531-t001:** Participant characteristics (*n* = 1376).

*M* ± *SD* or % (*n*)	
age (years)	45.64 ± 12.26
Sex female male	80.20 (1103) 19.80 (273)
civil status married single union divorced widow	49.80 (685) 21.90 (301) 16.10 (222) 10.30 (141) 1.90 (26)
education level primary secondary professional bachelor graduate master doctorate	0.90 (12) 6.40 (87) 2.00 (28) 3.80 (52) 57.80 (790) 25.20 (345) 3.90 (53)
subgroups general population nurses	54.30 (747) 45.70 (629)
sleep quality	5.36 ± 2.21
sleep quality above average below average	46.00 (633) 54.00 (743)
depressive symptoms	3.88 ± 2.42
anxiety	4.87 ± 2.51
irritability	4.83 ± 2.51
economic problems	3.21 ± 2.24
worries about the future	6.10 ± 2.46
sleep duration during weekdays (hours)	6.69 ± 1.72
sleep latency (minutes)	37.46 ± 37.55
sleep awakenings quality	5.51 ± 2.19
*M*: mean; *SD*: standard deviation.	

**Table 2 ijerph-20-05531-t002:** Variable descriptions included in the study (*n* = 1376).

Study Variables	Range	
sleep quality *	minimum = 1	maximum = 10
depressive symptoms	minimum = 1 (no depression)	maximum = 10 (high depression)
anxiety	minimum = 1 (no anxiety)	maximum = 10 (high anxiety)
irritability	minimum = 1 (no irritability)	maximum = 10 (high irritability)
economic problems	minimum = 1 (no economic problems)	maximum = 10 (high economic problems)
worries about the future	minimum = 1 (no worries)	maximum = 10 (high worries)
sleep duration weekdays (hours)	minimum = 2	maximum = 16
sleep latency (minutes)	minimum = 0	maximum = 240
sleep awakenings quality	minimum = 1	maximum = 10

* The variable was dichotomized taking as a reference the mean sleep quality (below average or above average).

**Table 3 ijerph-20-05531-t003:** Bivariate analysis of the variables in study (*n* = 1376).

	General Population 54.30% (*n* = 747)	Nurses 45.70% (*n* = 629)	
	*M* ± *SD* or % (*n*)		*p*
age (years) ^1^	49.32 ± 13.08	41.30 ± 9.56	<0.001
sex ^2^ female male	77.10 (576) * 22.90 (171)	* 83.78 (527) 16.22 (102)	0.002
sleep quality ^1^	5.69 ± 2.28	4.97 ± 2.04	<0.001
sleep quality above average below average	* 53.55 (400) 46.45 (347)	37.04 (233) * 62.96 (396)	<0.001
depressive symptoms ^1^	3.60 ± 2.37	4.21 ± 2.45	0.395
anxiety ^1^	4.61 ± 2.57	5.17 ± 2.41	0.018
irritability ^1^	4.38 ± 2.55	5.36 ± 2.36	0.007
economic problems ^1^	3.27 ± 2.31	3.13 ± 2.16	0.046
worries about the future ^1^	5.91 ± 2.48	6.34 ± 2.42	0.445
sleep duration weekdays ^1^	6.97 ± 1.64	6.33 ± 1.74	0.189
sleep latency ^1^	35.69 ± 39.01	39.58 ± 35.61	0.630
sleep awakenings quality ^1^	5.89 ± 2.25	5.08 ± 2.05	<0.001

^1^ Independent sample *t* test; ^2^ Chi-square. * Adjusted residuals > 1.96. *M*: mean; *SD*: standard deviation.

**Table 4 ijerph-20-05531-t004:** Logistic regression of sleep quality.

	Sleep Quality (Above Average)			
	OR (95% CI)	*p*	OR (95% CI) ^a^	*p*
subsample general population nurses	1.00 (ref.) 0.58 (0.44–0.77)	<0.001	1.00 (ref.) 0.51 (0.38–0.69)	<0.001
depressive symptoms	0.88 (0.80–0.96)	0.006	0.91 (0.83–1.00)	0.057
anxiety	0.93 (0.84–1.02)	0.107	0.93 (0.84–1.02)	0.119
irritability	0.94 (0.86–1.01)	0.104	0.91 (0.83–-0.99)	0.020
economic problems	0.95 (0.89–1.03)	0.202	0.96 (0.89–1.03)	0.262
worries about the future	0.90 (0.84–0.96)	0.002	0.89 (0.84–0.96)	0.001
sleep duration weekdays	1.03 (0.95–1.13)	0.469	1.04 (0.95–1.13)	0.411
sleep latency (minutes)	1.00 (1.00–1.00)	0.798	1.00 (1.00–1.00)	0.702
*Nagelkerke* = 0.206, *X*^2^(8) = 166.388, *p* < 0.001			*Nagelkerke* = 0.225, *X*^2^(10) = 181.478, *p* < 0.001	

OR: odds ratio; CI: confidence interval. ^a^ The results were adjusted for age and sex.

## Data Availability

The data presented in this study are available on request from the corresponding author. The data are not publicly available due to privacy/ethical restrictions.

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
