# Peer review of "Sleep Quality between Nurses and the General Population during the COVID-19 Pandemic in Portugal: What Are the Differences?"

_ijerph, 2023, doi:10.3390/ijerph20085531_

Round 1
Reviewer 1 Report
Reviewer comments and suggestions
The author of this study examined the differences between nurses and the general population regarding sleep quality and mental health status during the COVID-19 pandemic and identify the factors which could explain sleep quality during the COVID-19 pandemic. The study design include cross-sectional study in Portugal. Data were collected using an online survey platform during the first COVID-19 wave, from April to August 2020. Nurses presented worse sleep quality than the general population, as well as higher anxiety levels. Moreover, the study concluded that irritability and worries about the future are dimensions of anxiety that were associated with poor sleep quality during the COVID-19 pandemic.
Overall, the manuscript was well written. However, a few concerns/comments needed to be explained/modified.
- Line 79-81 Please explore the sentence with the explaining the cited references
- Line 82-85 Please avoid long sentences
- Line 92-94 The authors need to discuss the working hours during COVID to verify their points of poor sleep quality, only citing references would not matter. Please explore these studies with their presented data.
- Line 105-106 The issue is that authors merged many issues with a sentence, and try to make simple writing for the common reader of your MS.
- Line 109-110 Its already known and published several studies. Please highlight new findings using new statistical tools.
- How they define sleep quality, should be mentioned in the methods
- Line 252 The authors wrote several studies but were unable to justify those studies that needed a full explanation to enrich your own results
- Line 256-258 Some points needed to be discussed based on the author's approach to preventing the conditions.
- Check the journal style of references 1 and 2
Author Response
Dear reviewer,
Firstly, we would like to thank you very much for your relevant and insightful comments. We are sure they helped substantially improve our paper. Below you can find a point-by-point response to all your recommendations:
- Line 79-81 Please explore the sentence with the explaining the cited references.
As requested, we explored the sentence based on the cited references. We noted that one the cited references was not correct, so we also replaced it with the correct one.
- Line 82-85 Please avoid long sentences.
Our paper was edited and proofread by a professional English editing service prior to its submission. Nonetheless, we tried to simplify the sentence to make it shorter.
- Line 92-94 The authors need to discuss the working hours during COVID to verify their points of poor sleep quality, only citing references would not matter. Please explore these studies with their presented data.
We added some additional information, which was included in the systematic review we had cited, regarding the healthcare workers working hours during the COVID-19 pandemic and its implications to their mental health (considering, as it was already stated in the manuscript, there is a clear relationship between sleep quality and mental health).
4. Line 105-106 The issue is that authors merged many issues with a sentence, and try to make simple writing for the common reader of your MS.
We divided it in two different sentences, so too many issues are not mixed in the same sentence anymore. Perhaps, the more complex term in the sentence is “regression analysis”; however, we consider that term should be maintained, as it is quite relevant for a correct understanding of the sentence by the scientific community.
- Line 109-110 Its already known and published several studies. Please highlight new findings using new statistical tools.
Indeed, some studies have already examined whether there were differences between nurses and the general population regarding sleep quality and mental health status during the COVID-19 pandemic. Nonetheless, as we stated on the paper, few studies have compared the sleep quality of nurses and the general population in the same timeframe and using the same measures. Additionally, that information gives us the basis and serves as the starting point to explore the second (and perhaps the most relevant) aim of the paper: to identify which factors may explain sleep quality during the COVID-19 pandemic.
- How they define sleep quality, should be mentioned in the methods
The sleep quality was dichotomized above average and below average as mentioned in the “Methods” section.
- Line 252 The authors wrote several studies but were unable to justify those studies that needed a full explanation to enrich your own results
In response to this comment, we made the association of the results of the indicated studies with the results of our study.
- Line 256-258 Some points needed to be discussed based on the author's approach to preventing the conditions.
Thank you for your comment. We have improved this paragraph of the discussion according to your suggestion.
- Check the journal style of references 1 and 2
The references are in accordance with MDPI guidelines.
Reviewer 2 Report
First of all, I want to congratulate the team of authors for the work done in creating this manuscript and I want to thank you for the invitation to participate in this review.
The article "Sleep Quality between Nurses and the General Population during the COVID-19 Pandemic in Portugal: What are the Differences?" emphasizes the negative impact of the COVID-19 pandemic on the health status of nurses compared to the negative effects on the general population.
The research results identify the main risk factors that can affect the quality of sleep during the pandemic for nurses and the general population.
The topic addressed by this article is topical and of particular interest to top management and policymakers who may undertake policies and safeguards to protect healthcare personnel.
In the Introduction section, a complete review of the specialized literature on the research topic with relevant and up-to-date citations from the last 5 years is carried out.
The methods section lacks information such as study design, inclusion/exclusion criteria, sample size calculation, outcome measures, statistical analyses, and so on. Authors should write this section according to the study design (I suggest using the STORBE reporting system).
I also recommend that the author mention what were the main measures that were taken to reduce the phenomenon of bias.
I think it would be useful to detail the data collection methodology: how it was done, what type of tool you used, how the data collection tool was developed, and how the data collection questionnaire was distributed to the study population. How authors assessed the validity and reliability of the questionnaires?
Please present what were the main measures to fulfil the ethical requirements of the research (for example consent to participate in the study, how it was brought to the knowledge of the participants, what measures were taken to preserve the confidentiality of the data, etc.:
Results
It would be useful to find out, if possible, what is the field of activity of the nurses included in the study - were the nurses who worked in hospitals, outpatient medical units, or residential centres? Depending on the workload and case history, the negative impact on health could be different.
The tables are adequate and appropriately display the data, being easy to interpret and understand.
Author Response
Dear reviewer,
Firstly, we would like to thank you very much for your relevant and insightful comments.
We are sure they helped substantially improve our paper. Below you can find a point-
by-point response to all your recommendations:
First of all, I want to congratulate the team of authors for the work done in
creating this manuscript and I want to thank you for the invitation to participate
in this review. The article "Sleep Quality between Nurses and the General
Population during the COVID-19 Pandemic in Portugal: What are the
Differences?" emphasizes the negative impact of the COVID-19 pandemic on the
health status of nurses compared to the negative effects on the general
population. The research results identify the main risk factors that can affect the
quality of sleep during the pandemic for nurses and the general population. The
topic addressed by this article is topical and of particular interest to top
management and policymakers who may undertake policies and safeguards to
protect healthcare personnel.
Thank you for the comments.
In the Introduction section, a complete review of the specialized literature on the
research topic with relevant and up-to-date citations from the last 5 years is
carried out.
Thank you for the comments.
The methods section lacks information such as study design,
inclusion/exclusion criteria, sample size calculation, outcome measures,
statistical analyses, and so on. Authors should write this section according to
the study design (I suggest using the STORBE reporting system).
Thank you for the comments. The “Methods” section was revised according to your
comments. We added study design, sampling method and the outcome measures were
improved / clarified.
I also recommend that the author mention what were the main measures that
were taken to reduce the phenomenon of bias.
Thank you for the comments. To reduce bias we performed the following main actions:
1) compared the sociodemographic characteristics of the different subgroup in study; 2)
before implementing the survey, it was applied to a sub-sample for checking
understanding and relevance of each question.
I think it would be useful to detail the data collection methodology: how it was
done, what type of tool you used, how the data collection tool was developed,
and how the data collection questionnaire was distributed to the study
population. How authors assessed the validity and reliability of the
questionnaires?
Thank you for the comments. The “Methods” section was revised according to your
comments. To ensure that the questionnaires were adequate they were revised by a
team of peers and before implementing the survey they were applied to a sub-sample
for checking understanding and relevance of each question.
Please present what were the main measures to fulfil the ethical requirements of
the research (for example consent to participate in the study, how it was brought
to the knowledge of the participants, what measures were taken to preserve the
confidentiality of the data, etc.:
Thank you for the comments. The study was approved by an Ethical Committee and
the consent of the participants was obtained before starting to fill out the questionnaire.
Also, before starting the questionnaire and asking for consent, the objectives and
procedures of the study were present to each participant. To preserve confidentiality of
the data, surveys were conducted anonymously.
Results
It would be useful to find out, if possible, what is the field of activity of the
nurses included in the study - were the nurses who worked in hospitals,
outpatient medical units, or residential centres? Depending on the workload and
case history, the negative impact on health could be different.
Thank you for the comments. The suggestion is interesting but it wasn´t the aim of this
study. We intended to analyse nurses’ sleep quality as an ‘overall’ population. However,
is an interesting topic for future research.
The tables are adequate and appropriately display the data, being easy to
interpret and understand.
Thank you for the comments.